# Geometry-Aware Recurrent Neural Networks for Active Visual Recognition

**Ricson Cheng**, **Ziyan Wang**, **Katerina Fragkiadaki**
Carnegie Mellon University
Pittsburgh, PA 15213
{ricsonc,ziyanw1}@andrew.cmu.edu, katef@cs.cmu.edu

## Abstract

We present recurrent geometry-aware neural networks that integrate visual information across multiple views of a scene into 3D latent feature tensors, while maintaining an one-to-one mapping between 3D physical locations in the world scene and latent feature locations. Object detection, object segmentation, and 3D reconstruction is then carried out directly using the constructed 3D feature memory, as opposed to any of the input 2D images. The proposed models are equipped with differentiable egomotion-aware feature warping and (learned) depth-aware unprojection operations to achieve geometrically consistent mapping between the features in the input frame and the constructed latent model of the scene. We empirically show the proposed model generalizes much better than geometry-unaware LSTM/GRU networks, especially under the presence of multiple objects and cross-object occlusions. Combined with active view selection policies, our model learns to select informative viewpoints to integrate information from by "undoing" cross-object occlusions, seamlessly combining geometry with learning from experience.

## 1 Introduction

Cross-object occlusions remain an important source of failures for current state-of-the-art object detectors [29], which, despite their formidable performance increase in recent years, still carry the biases and idiosyncrasies of the data they were trained on [16]: static images from Imagenet and COCO datasets. Thus, detectors do well on viewpoints that human photographers tend to prefer, but may fail on images (non skillfully) captured by robotic agents, that often feature severe cross-object occlusions and viewpoints which are unusual for humans. Robotic agents on the other hand do not abide by the constraints of *passive* vision: they can actively *choose where to look* in order to recover from occlusions and difficult viewpoints.

Actively selecting camera views for "undoing" occlusions and recovering missing information has been identified as an important field of research since as early as 1980's, under the name *active vision* [1]. It has been argued that without such active view selection, no formal performance guarantees can be provided for visual recognition algorithms [32]. Yet, 1980's active vision was not equipped with deep neural detectors, memory modules, or view selection policies, and often attempted tasks and imagery that would appear elementary with current detectors, even from a single camera view. There has been recent methods that have attempted to resurrect the active vision paradigm [23, 9, 4, 18]. Paradoxically, they either focus on sequential processing of selected visual glimpses from a *passive static* image [23, 9, 4], and thus they are not able to "undo" occlusions, or they consider 3D reconstruction of *isolated* objects [18, 43, 20], while one can argue that isolating objects from clutter is precisely the premise of having an active camera.

We propose **geometry-aware recurrent networks and loss functions for active object detection, segmentation and 3D reconstruction in cluttered scenes**, following the old active vision premise. Our main contribution is a network architecture that accumulates information across camera views into a 3D feature map of the scene, in a geometrically consistent manner, so that information regarding the same 3D physical point across views is placed nearby in the tensor. **Object segmentation, classification and 3D reconstruction is directly solved for from the output of the 3D latent feature tensor, as opposed to in 2D input images**. This is drastically different from the paradigm of tracking by detection [40] where we need to commit to object detections in each image and post-correspond them in time, and may suffer from early erroneous decisions. Our model instead accumulates visual information across multiple views in a space-aware latent space, before detecting the objects, and thus detection performance strictly improves with more views. Projecting the detected 3D object segments generates *amodal* boxes and segmentations in the (potentially heavily) occluded input 2D views.

In a nutshell, our method works as follows. At each time step, an active observer selects a nearby camera viewpoint. Our model then "unprojects" RGB, (learned) depth and object foreground mask into a 3D feature tensor, which is rotated to match the pose of the first camera view, using the relative egomotion, which is assumed known. The warped feature tensor then updates a 3D latent state map of a convolutional Gated Recurrent Unit (GRU) layer. Geometric consistency ensures that *information from 2D projections of the same 3D physical point are placed nearby in the 3D feature memory tensor*, a key for generalization. GRU convolutional filters can then "see" information regarding the same 3D concept in their vicinity. They transform it into higher level features optimized for object instance segmentation, classification and voxel occupancy prediction, directly in 3D. Our geometry-aware recurrent network is trained end-to-end in a differentiable manner, and our active view selection is trained with reinforcement rewarded by state estimation accuracy at each time step.

We empirically show the proposed 3D recurrent network outperforms by a large margin 2D recurrent network alternatives, convolutional or not, which do not effectively generalize in cluttered scenes due to the combinatorial explosion of occlusion configurations. Our view selection policy outperforms naive or random view selection alternatives. The proposed method marries pure geometric localization and 3D pointcloud mapping [31] with feature learning from experience, while being robust to mistakes of depth and egomotion estimation. In summary our contributions are:

- Introducing the problem of active state estimation, namely, selecting camera views for jointly optimizing object instance segmentation, classification and 3D reconstruction, a problem very relevant for robotic computer vision.
- Proposing a geometry-aware recurrent neural network that accumulates feature information directly in 3D. The proposed model can be used for both passive as well as actively collected video sequences.
- Conducting extensive experiments and ablations against alternative recurrent architectures in both "easy" and "hard" environments.

Our codes will be made available at `ricsonc.github.io/geometryaware`.

## 2  Related Work

**Active vision**    An active vision system is one that can manipulate the viewpoint of the camera(s) in order to explore the environment and get better information from it [3, 39]. Active visual recognition attracted attention as early as 1980's. Aloimonos *et al.* [1] suggested many problems such as shape from contour and structure from motion which, while underconstrained for a passive observer, can easily be solved by an active one [7, 30]. Active control of the camera viewpoint also helps in focusing computational resources on the relevant elements of the scene [35]. The importance of active data collection for sensorimotor coordination has been highlighted in the famous experiment of Held and Hein in 1963 [14], involving two kittens engaged soon after birth to move and see in a carousel apparatus. The active kitten that could move freely was able to develop healthy visual perception, but the passively moved around kitten suffered significant deficits. Despite its importance, active vision was put aside potentially due to the slow development of autonomous robotics agents, necessary for the realization of its full potential. Recently, the active vision paradigm has been resurrected by approaches that perform selective attention [28] and selective sequential image glimpse processing in

order to accelerate object localization and recognition in 2D static images [23, 9, 4]. Fewer methods have addressed camera view selection for a moving observer [2, 45, 22], e.g. for object detection [17], for object recognition [19], for navigation [24, 11] or for object 3d reconstruction, in synthetic environments [18]. Since we cannot backpropagate through viewpoint selection, reinforcement learning methods have often been used to train view or glimpse selection policies. Many of the aforementioned approaches use some type of recurrent network [23, 9, 4, 25, 20] that learns to integrate information across views or glimpses. Our work proposes a particular architecture for such recurrent network that takes into account the egomotion of the observer.

**Deep geometry**   There has recently been great interest in integrating learning and geometry for single view 3D object reconstruction [36, 41], depth and egomotion estimation from pairs of frames [38, 44], depth estimation from stereo images [8], estimating 3D locations of human keypoints from 2D keypoint heatmaps [42, 37]. Many of those works use neural network architectures equipped with some form of differentiable camera projection, so that 3D desired estimates can be supervised directly using 2D quantities. They do not accumulate information across multiple camera viewpoints. For example, Tulsiani *et al.* [36], Wu *et al.* [41] and Zhou *et al.*[44] use a single image frame as input to predict 3D reconstruction of a single object, or a 2D depth map of the entire scene, though they use multiple views to obtain supervision in the form of depth re-projection error. Works that use reprojection error as self-supervision at test-time [42, 37, 38, 44, 36, 41] cannot operate in an online, interactive mode since backpropagating error gradients takes time. Learnt stereo machines (LSM) [20] integrates RGB information along sequences of random camera viewpoints in a 3D feature memory tensor, in an egomotion stabilized way, similar to our method. We compare our model to LSM [20] in Section 4 and show that using learning-based depth estimation and early feature unprojection generalizes better in the presence of multiple objects that depth-unaware late unprojection, used in LSM [20]. 3D ShapeNets [43] uses a deep generative model that given a depth map as input predicts a 3D occupancy voxel grid, and has an unprojection operation similar to the one we propose. It is trained using contrastive divergence while our model is a simple recurrent discriminative network with straightforward training. Both LSM [20] and 3D ShapeNets [43] consider the problem of single object 3D reconstruction. Our work instead considers segmentation, classification and 3D reconstruction of multiple objects in clutter. Pointnets [27] are neural network architectures that operate directly on 3D pointcloud input, as opposed to pixel grids. They do not accumulate information across multiple views.

Purely geometric approaches, such as Simultaneous Localization and Mapping methods [31, 21] build a 3D map of the scene while estimating the motion of the observer. Our method builds a feature map instead, which captures both the geometry and the semantics of the scene. Each point of the map is a feature vector as opposed to an occupancy label. SLAM method need to see the scene from all viewpoints in order to reconstruct it, they cannot map the invisible. Our model instead learns to fill in missing views from experience.

MapNet [15], IQA [10] and Neural Map [26] construct 2D birdview maps of the scene by taking into account the egomotion of the observer, similar to our method. MapNet further estimates the egomotion, as opposed to assuming it known as the rest of the methods, including ours. They consider the tasks of navigation [15, 26] and interactive question answering [10], while we consider the task of object detection, segmentation and classification. In IQA, objects are detected in each frame and detections are aggregated in the birdview map, whereas we detect objects using the feature map directly as input. If an object is too heavily occluded to be reliably detected in any single view, it can still be inferred given the integrated information across all views.

## 3   Active vision with geometry-aware recurrent neural networks

We consider an active observer, e.g., a robotic agent, equipped with a perspective camera model. At each timestep, the agent selects a nearby camera viewpoint. The RGB, depth and foreground mask of the selected view are then "unprojected" into a 3D tensor and transformed in order to compensate for the egomotion of the observer, namely, the translation and rotation between the current and previous views. The oriented tensor is then fed to a recurrent neural network that builds a latent model of the scene, as illustrated in Figure 1. We describe our geometry-aware recurrent model and viewpoint selection policy right below.

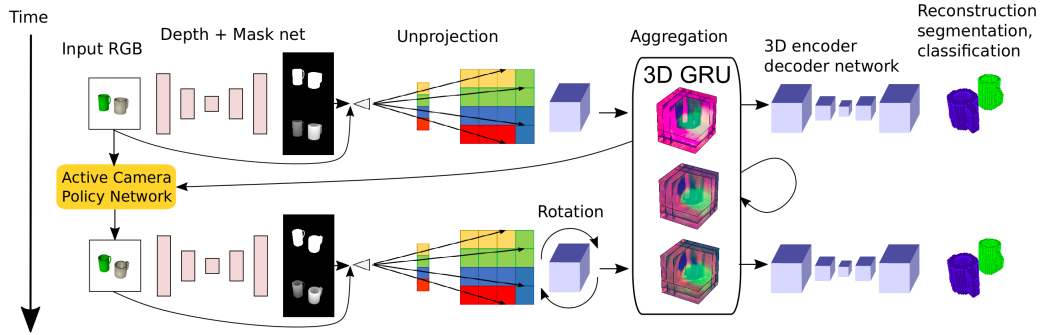

Figure 1: **Active vision with geometry-aware recurrent neural networks.** Our active agent sequentially accumulates visual information in a 3D geometrically-consistent feature map of the visual scene. At each frame, the agent selects a nearby camera view, conditioned on the 3D feature map of the scene thus far, and the current RGB image. It predicts a corresponding 2D depth map and foreground object mask. RGB, depth and segmentation maps are unprojected in corresponding 3D feature tensors, mapped using the relative egomotion to the coordinate frame of the cumulative 3D feature map built thus far, and update a 3D convolutional GRU memory. We visualize 3 channels of the memory. The output of the 3D GRU memory is then mapped via a 3D encoder-decoder network to segmentation embeddings, object category probabilities, and occupancy for every voxel in the 3D grid.

## 3.1 Geometry-aware recurrent neural network

Our memory model is a recurrent neural network whose latent state corresponds to a 3D feature map of the visual scene. The memory map is updated with each new camera view in a geometrically consistent manner, so that information from 2D pixel projections that correspond to the same 3D physical point end up nearby in the tensor. This permits convolutional operations to have a correspondent input across views, as opposed to varying with the motion of the observer.

**Unprojection**   In an RGB image, we predict the corresponding 2D depth and object foreground map using a 2D convolutional encoder-decoder network with skip-connections. We then *unproject* those to fill in an initial 3D feature tensor as follows:

For each feature voxel $V_{i,j,k}$ in the 3D tensor we compute the pixel location $(x, y)$ which it projects onto, from the current camera view. Then, $V_{i,j,k}$ is filled with the bilinearly interpolated RGB, depth and mask at $(x, y)$. All voxels lying along the same ray shot from the camera will be filled with nearly the same RGB, depth and mask, although in practice, since the centroids of two voxels are rarely on the exact same camera ray, the features will differ slightly.

We further compute (i) a grid of depth values, so that $V_{i,j,k} = k$ and (ii) a binary voxel occupancy grid that contains the thin shell of voxels directly visible from the current camera view. We compute this by filling all voxels whose unprojected depth value equals the grid depth value. Thus, our unprojection operation results in an initial feature tensor $\mathcal{M}^{64 \times 64 \times 64 \times 7}$: 3 channels for RGB, 2 channels for depth and mask, 1 channel for the surface occupancy, and 1 channel for the grid of depth values. We visualize the unprojection operation for one horizontal slice and one channel of the 3D feature tensor in Figure 1.

**Recurrent memory update**   In order for the 2D projections of the same physical point to end up closeby in our 3D memory tensor, in each view, we rotate unprojected feature tensors to the orientation of the first camera view. This requires only relative camera pose (egomotion). We will assume ground-truth egomotion information throughout the paper since an active observer does have access to its approximate egomotion, and leave egomotion estimation as future work. In Section 4, we quantify how robust our recurrent model is to inaccurate egomotion.

Once the feature tensor has been properly oriented, we feed it as input to a 3D convolutional Gated Recurrent Unit [6] layer. The operation of the GRU memory layer update can be written as:

$$h_{t+1} = u_t \circ h_t + (1 - u_t) \circ \tanh(\text{conv3d}([x_t, r_t \circ h_t], W_h)) \tag{1}$$
$$u_t, r_t = \sigma(\text{conv3d}([x_t, h_t], [W_u, W_r])) \tag{2}$$

where $u_t$ and $r_t$ are the GRU update gate and reset gate respectively. We experimented with max and average pooling operations for such memory update, but found that the GRU memory update worked best.

**Object reconstruction, instance segmentation and classification**   Our model is trained in a supervised manner to predict object instance segmentation, object classification and 3D object reconstruction (voxel occupancy) using 3D groundtruth object bounding boxes, masks, and occupancy voxel grids available in simulator environments.

The output of the 3D GRU memory is fed into a 3D convolutional encoder/decoder network with skip-connections to produce the final set of outputs: (i) a 3D sigmoid output which predicts voxel occupancy, (ii) a 3D segmentation embedding feature tensor, and (iii) a multiclass softmax output at every voxel for category labelling. The segmentation embeddings and classification outputs have half the resolution of the occupancy grid.

We train for 3D reconstruction (voxel occupancy) using a standard binary cross-entropy loss. We train for object instance segmentation by learning voxel segmentation embeddings [13]. Clustering using the learnt embedding distances provides a set of voxel clusters, ideally, each one capturing an object in the scene. We use metric learning and a standard contrastive loss [12], that brings segmentation embeddings of voxels of the same object instance close, and pushes segmentation embeddings of different object instances (and the empty space) apart. During training, we sample the same number of voxel examples from each object instance to keep the loss balanced, not proportional to the volume of each object. We convert the voxel feature embeddings into a voxel segmentation map as follows: We discard voxel whose predicted occupancy is less than 0.5. We cluster the remaining voxels using $k$-means, with $k$ set to the number of objects in the scene. When the number of objects is unknown, we oversegment the scene using a large $k$, as shown in Figure 2. We then iteratively merge clusters whose ratio of the number of adjacent voxels to the volume of the larger of the two clusters, $R$ is the largest. We stop merging when $R < 1.5$. In our scenario, such clustering is much facilitated by the fact that objects live in 3D and free space effectively separates the different objects, preventing clusters to "leak" across different object instances.

For each voxel cluster, we compute the average of the logits predicted for the voxels in that group, and use that as the predicted class of the object. We train with cross-entropy loss between the average logit and the ground truth class label of the (groundtruth) shape in the scene which has the highest Intersection over Union (IoU) with each voxel cluster.

## 3.2 Learning a view selection policy

We train a view selection policy to predict at each timestep a distribution over eight adjacent camera views in the neighborhood of the current one. Our policy network is a convolutional neural network with two branches, one 2D and one 3D. The input to the 3D branch is the output of the GRU memory layer, and the input to the 2D branch is the current RGB image. They are both mapped, using 3D and 2D convolutions respectively, to a final 1D vector after linearization and concatenation. A fully connected layer with softmax activation produces the final categorical distribution over 8 possible directions. Description of the detailed architecture of the view selection policy network is included in the Appendix.

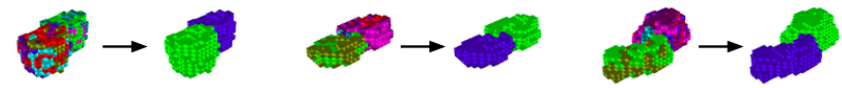

Figure 2: **Instance segmentation** The left image in each of the 3 pairs shows the oversegmented scene using $k = 8$. We iteratively merge the clusters as described in the text to obtain the final segmentation result.

We train our policy network with REINFORCE [34] with a Monte Carlo baseline. We tried more advanced Actor-Critic methods, but we did not find a difference empirically. The policy $\pi(a_t|s_t, \theta)$ is iteratively updated with gradient ascent to maximize the expectation of total reward. The policy gradient is $g(s) = E_t[\nabla_\theta \log(\pi_\theta(a_t|s_t, \theta))R_t]$ where $R_t$ is the reward of step $t$. $E_t[\cdot]$ is the empirical expectation of the average over a finite number of sampled time steps. Though our agent solves for object segmentation, classification and 3D reconstruction, we found it adequate to provide reconstruction-driven rewards $R_t$, as the *increase* in Intesection over Union (IoU) of the discretized voxel occupancy from each view to the next.

## 4 Experiments

We test our models in two types of simulated environments: i) scenes we create using synthetic 3D object models from ShapeNet [5]. The generated scenes contain single or multiple objects on a table surface. ii) scenes from the SUNCG [33] dataset, sampled around an object of interest. Each 3D scene is rendered from a viewing sphere which has $3 \times 18$ possible views with 3 camera elevations $(20°, 40°, 60°)$ and 18 azimuths $(0°, 20°, \ldots, 340°)$. The agent is allowed to move to one of the 8 adjacent views at every time step. We voxelize each 3D scene at a resolution of $64 \times 64 \times 64$.

We pretrain our geometry-aware recurrent neural network using randomly selected view sequences so that it does not overfit to the biases of the view selection policy. We then train our view selection policy while the weights of the recurrent neural network are kept fixed.

**Multi-view reconstruction of single objects**   We compare the performance of the geometry-aware recurrent network against alternatives on the task of 3D reconstruction of single objects using *randomly selected* camera views (no active vision). We consider three object categories from ShapeNet: chairs, cars, and airplanes. We split the data into training, validation, and test according to [5]. We train a single 3d reconstruction model for all categories using supervision from groundtruth 3D voxel occupancy. We compare the proposed geometry-aware recurrent net *ours* against the following baselines:

1. *1D-LSTM*: an LSTM network, similar to the one used in [41, 18]. In each frame, given RGB and depth as input, it produces through a 2D convolutional encoder a feature embedding vector by flattening the last 2D convolutional feature map. The embedding vector is fed into a (fully-connected) LSTM layer and the output of the LSTM is decoded into a 3D occupancy grid through a series of 3D convolutions. We detail its architecture in the Appendix.

2. Learnt Stereo Machines (*LSM*) of [20]. This model aggregates information using a 3D GRU layer, similar to ours, by unprojecting convolutional features extracted from RGB images using *absolute* groundtruth camera poses. Absolute pose may be hard to obtain and it may not be meaningful in the presence of multiple objects, our method only requires relative pose. They use a late unprojection as opposed to our proposed early depth-aware unprojection. We use the publicly available code of [20] and retrain their model for our setup.

3. *LSMdepth*. An LSM alternative that uses depth as an additional input channel and performs late unprojection on features extracted from the concatenation of depth and RGB.

All methods are trained and tested on sequences of randomly selected nearby camera views.

We report Intersection over Union (IoU) between the predicted and groundtruth occupancy grids in each of four camera views for our model and baselines in the test set in Table 1. More visualization results are included in the supplementary material. The proposed geometry-aware recurrent neural network outperforms the baselines. The LSM outperforms our method on the first input view. However, after aggregating information from more views, our method consistently outperforms all alternatives.

For SUNCG, we similarly test our proposed model on 3D reconstruction of (usually single) objects using randomly selected camera views as input. Intersection over Union (IoU) between the predicted and groundtruth voxel occupancy grids in each of four consecutive camera views for our model and LSTM baseline are shown in Table 2. Qualitative results are included in the Appendix.

**Active view selection**   We evaluate the performance of our active view selection policy on 3D reconstruction of single and multiple objects. At each frame, our active observer learns to select an

|  | single object | | | | multi-objects | | | |
| --- | --- | --- | --- | --- | --- | --- | --- | --- |
|  | view-1 | view-2 | view-3 | view-4 | view-1 | view-2 | view-3 | view-4 |
| 1D-LSTM | 0.57 | 0.59 | 0.60 | 0.60 | 0.11 | 0.15 | 0.17 | 0.20 |
| LSM | 0.63 | 0.66 | 0.68 | 0.69 | 0.43 | 0.47 | 0.51 | 0.53 |
| LSM+gt depth | **0.65** | 0.68 | 0.69 | 0.70 | **0.48** | 0.51 | 0.54 | 0.56 |
| ours+gt depth | 0.55 | **0.69** | **0.72** | **0.73** | 0.47 | **0.58** | **0.62** | **0.64** |
| ours+learnt depth | - | - | - | - | 0.45 | 0.56 | 0.60 | 0.62 |

Table 1: **3D voxel occupancy prediction using randomly selected camera views.** We show Intersection over Union (IoU) between the prediction and groundtruth 3D voxel grids. The performance of our model improves with more views. Using estimated depth as input does slightly worse than using groundtruth depth.

|  | view-1 | view-2 | view-3 | view-4 |
| --- | --- | --- | --- | --- |
| 1D-LSTM | 0.12 | 0.16 | 0.16 | 0.18 |
| ours | 0.24 | 0.28 | 0.31 | 0.32 |

Table 2: **3D voxel occupancy prediction using randomly selected camera views.** We show Intersection over Union (IoU) between the predicted and groundtruth 3D voxel occupancies on SUNCG scenes.

*adjacent* —as opposed to random [18]— view to jump to, in an attempt to mimic realistic deployment conditions on a robotic agent. We compare our learned view selection policy with four baselines:

1. *random*, which randomly selects neighboring views to visit, and

2. *oneway*, which moves the camera along the same direction at each timestep so that the agent maximizes the horizontal or vertical baseline.

3. *1-step greedy*, similar to the one used in [43], which always selects the next view by maximizing the immediate increase in IoU for a single step as opposed to a finite horizon.

4. *oracle*, which selects the best among all possible view trajectories through exhaustive search. Since it is too expensive to try out all possible four-view-long trajectories, we sample 100 four-view-long trajectories and take the best as the oracle trajectory. This serves as an upper-bound for our view selection policy.

We show qualitative results in Figure 3. For quantitative results on *random* and *oneway* policy, please see supplementary materials. The learned view selection outperforms the baselines for both single and multiple objects present. It better reconstructs detailed object parts, like the arm of a chair or the inside part of a mug. Compared to the *oneway* policy, the *active* policy select views that "undo" occlusions and recover most of the object.

We trained our policy using three random seeds and report the mean and variance.

The percent increase in IoU between predicted and groundtruth voxel occupanices over single-view predictions is reported in Table 3. We can see that our learned policy is worse than 1-step greedy policy in the first view, but outperforms it as more views become available. This is not surprising, as our learned policy was trained to select views that maximize the cumulative future rewards over a time horizon of four views. Qualitative results of the view selection policy are included in the supplementary materials.

**Multi-view, multi-object segmentation, classification and 3D reconstruction** No previous work has reported results on the task of object detection, segmentation and 3D reconstruction **under the presence of multiple objects occluding each other.** An important contribution of this paper is to address joint object instance segmentation, object category prediction and 3D reconstruction while accumulating information across multiple views in order to reveal occluded objects. We choose objects from four different categories, helmet, mug, bowl, and camera, that are similar in size. We place two objects in each scene between 0.25 and 0.35 units from the origin. The line passing through the centroid of both objects also goes through the origin. The scene is randomly rotated between $-90°$ and $90°$ degrees. Although we only generated 332 scenes, our network is able to perform and generalize well. We uses $70\%$ of the scenes for training and the rest for testing.

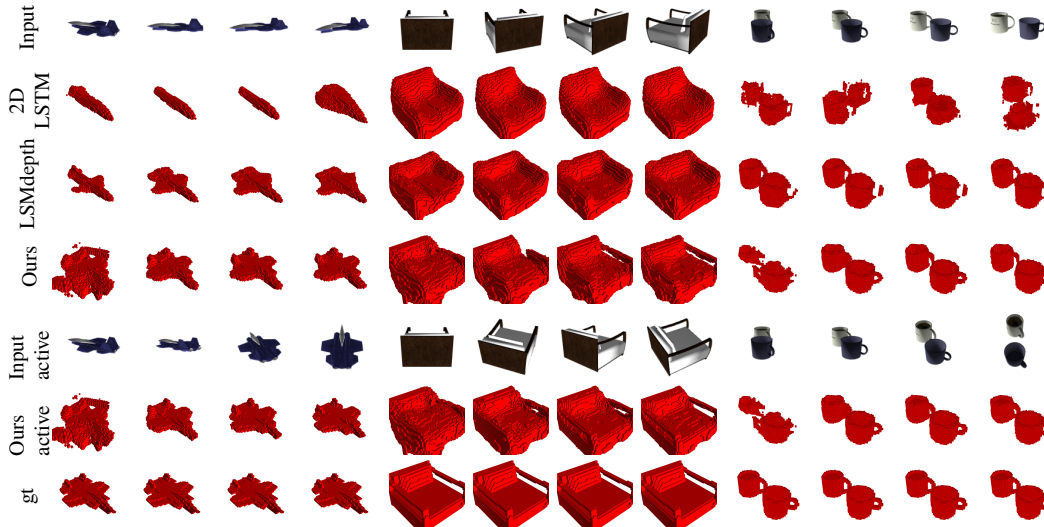

Figure 3: **3D reconstruction results** on single and multiple objects using different methods. Note the ability of our model to model the thin armrest of the chair without blending it into the body.

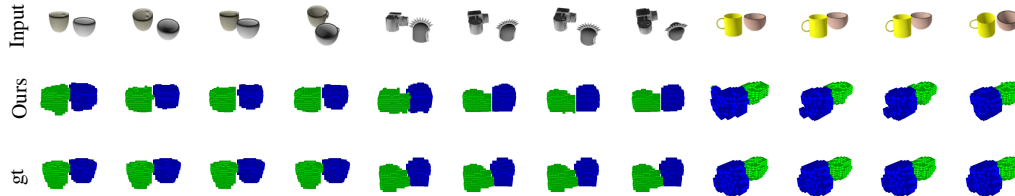

Figure 4: 3D segmentation on multi-object scenes.

We retrain the model of [20] using our proposed state estimation loss functions and with absolute groundtruth camera poses as a comparison. We report multi-object 3D reconstruction (voxel occupancy prediction) in Table 1 and classification and segmentation in Table 4. For 3D reconstruction, learned depth always results in lower reconstruction IoU in each view. The proposed architecture outperforms LSM, and depth input always helps, even if it is estimated by a network as opposed to groundtruth. This result is encouraging given the improvement of depth sensors currently available that the proposed geometry-aware architecture can maximally take advantage of. For object segmentation, we compute the 3D IoU using the optimal matching between the segmented 3D voxel groups recovered from $k$-means and the groundtruth 3D object voxel grids. We report the average IoU across all objects in a scene. We also report classification accuracy. Table 4 shows the quantitative results, and Figure 4 and 5 shows visualizations of the instance segmentation and classification.

|  | view-1 | view-2 | view-3 | view-4 |
|---|---|---|---|---|
|  | single object | | | |
| 1-step greedy | 0% | 29.1% | 32.7% | 33.3% |
| Active | 0% | 27.7%±3.23% | 27.7%±3.23% | 36.6%±1.17% |
| Oracle | 0% | 32.7% | 38.1% | 38.1% |
|  | multiple objects | | | |
| 1-step greedy | 0% | 36.2% | 40.4% | 40.4% |
| Active | 0% | 31.3%±2.26% | 39.0%±1.99% | 41.8%±2.36% |
| Oracle | 0% | 29.8% | 42.6% | 44.7% |

Table 3: Percent increase in Intersection over Union (IoU) between 3D reconstructions and groundtruth over the single-view reconstruction. The first view is the same for all policies, so the IoU for the first view is the same.

|                         | view-1 | view-2 | view-3 | view-4 |
|-------------------------|--------|--------|--------|--------|
| 3D voxel occupancy IoU  | 0.54   | 0.63   | 0.64   | 0.65   |
| 3D segmentation IoU     | 0.60   | 0.69   | 0.70   | 0.71   |
| Classification accuracy | 0.56   | 0.83   | 0.83   | 0.83   |

Table 4: **3D object segmentation, classification and 3D reconstruction using two objects per scene and four object categories.** As expected, both reconstruction and segmentation IoU increase with more views. The four object categories are roughly balanced and always guessing the most common category yields 32% accuracy.

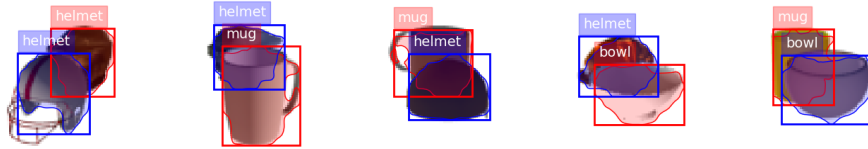

Figure 5: **Object segmentation under occlusions**. We are able to project the segmented 3D voxels and obtain amodal bounding boxes and segmentation masks, even in the presence of heavy occlusions. We show segmentations specifically from the most heavily occluded camera views.

**Sensitivity to egomotion estimation errors** Our reconstruction method uses the relative camera pose between views in order to align the unprojected feature tensors, before feeding them to a 3D GRU layer. However, there may be egomotion estimation errors in a realistic scenario. We test the robustness of our model to such errors by adding random noise to azimuth and elevation. We consider noise distributed normally with $\mu = 0$ and $\sigma = 5$ degrees for both the azimuth and elevation. We test noise which is independent for each view, and also noise that accumulates across views, so that the total error at view $i$ is the error at view $i - 1$ plus the additional error at view $i$. We test two variants of our method: one trained without noisy egomotion, and one trained with (independent) noisy egomotion. We show quantitative results in Table 5.

|                             | Trained without noise | Trained with noise |
|-----------------------------|-----------------------|--------------------|
| groundtruth egomotion       | **0.64**              | 0.63               |
| Independent egomotion noise | 0.56                  | **0.59**           |
| Accumulating egomotion noise| 0.56                  | **0.58**           |

Table 5: **3D reconstruction under noisy egomotion** for the chair category. We show IoU for the different test-time scenarios listed in the left column. The performance of our model degrades gracefully with noise in egomotion estimation. Moreover, when trained with such noise, it become more resistant to such errors.

## 5  Discussion - Future Work

We presented a method for active object detection, segmentation and 3D reconstruction in cluttered static scenes, that selects camera viewpoints and integrates information into a 3D feature memory map, using geometry-aware recurrent neural updates. Object detection, segmentation and 3D reconstruction are directly predicted from the constructed 3D feature map, as opposed to any single 2D view. As a result, object delineation seamlessly improves with more views, with no need to commit to early erroneous object detections, which are hard to correct once made. We showed the proposed model generalizes better than geometry-unaware LSTMs/GRUs, especially in cluttered scenes. The main contributing factor for the effectiveness of the proposed method is the representation: 3D space does not suffer from instantaneous occlusions or foreshortenings. It is a stable space for information integration.

The current framework has the following limitations: (i) it assumes ground-truth egomotion, (ii) it consumes lots of GPU memory for maintaining the full scene tensor, (iii) it cannot handle independently moving objects, and (iv) it requires 3D groundtruth for object detection, which is very expensive to obtain. Addressing the above limitations are interesting directions for future work.

## Footnotes

*Indicates equal contribution

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
