[Supplementary Material]

# Geometry-Aware Recurrent Neural Networks for Active Visual Recognition – Supplementary material

## 1 Notation

We denote conv2d(fw,fh,ch,st) as a 2D convolutional layer with ch filters of size (fw,fh) and a stride of st, conv3d(fw,fh,fl,ch,st) as a 3D convolutional layer with ch filters of size (fw,fh,fl) and stride of st, deconv2d(fw,fh,ch,st) as 2D deconvolutional layer with ch filters of size (fw,fh) and a stride of st, deconv3d(fw,fh,fl,ch,st) as 3D deconvolutional layer with ch filters of size (fw,fh,fl) and a stride of st, flatten(·) as a layer which flattens the input and maintaining the batch size, and fc(x) as fully connected layer with x hidden units.

## 2 Policy Network

Figure 1: The policy network we use for active view selection policy.

The architecture of policy network we used for active view selection policy is shown in Figure 1. The network has two inputs: RGB image of the current view and the aggregated feature voxel which is the output of the GRU memory layer. Then 2D convolutional layers and 3D convolutional layers are used to extract features from the RGB channel and the feature voxel channel respectively. The features extracted from two channels are flattened and concatenated along the last axis. A MLP is used to classify the concatenated feature and output a categorical distribution over 8 possible directions. The architecture of policy network in show in Table 1. After each layer, a leaky relu activation with slope ratio of $\alpha = 0.2$ and a batch normalization layer is used.

## 3 Model Architecture

We detail the architecture of the LSTM baseline, the depthnet, and our 3D reconstruction model.

| RGB branch | 3D feature branch |
|---|---|
| conv2d(3,3,3,32,2) | conv3d(3,3,3,16,2) |
| conv2d(3,3,3,32,2) | conv3d(3,3,3,32,2) |
| conv2d(3,3,3,64,2) | conv3d(3,3,3,32,2) |
| conv2d(3,3,3,64,2) | conv3d(3,3,3,64,2) |
| conv2d(3,3,3,128,2) | conv3d(3,3,3,128,2) |
| flatten | |
| fc(4096) | |
| fc(4096) | |
| fc(8) | |
| softmax | |

Table 1: Architecture of policy network

**LSTM** The LSTM baseline contains three parts: An encoder which encodes the input image or feature into a feature vector, a fuser which fuses the feature vector into a latent feature vector, an aggregator which aggregates on the latent feature given by fuser, and a 3D decoder which maps the aggregated latent features into a 3D voxel occupancy prediction. The architectures of the encoders (2D input and 1D input), fuser, and 3D decoder are shown in Table 2.

For RGB and depth input, we use the same image encoder for 2D input. For camera pose input, we input the azimuth and elevation of camera into pose encoder for. Then we concatenate the features extracted from RGB, depth and camera pose using encoders along the last axis. The fuser takes the concatenated features as input and output a fused feature.

Then an LSTM with 4096 hidden units is used to aggregate on the fused feature. And a 3D decoder maps the fused feature into a voxel occupancy prediction.

| Image Encoder(2D input) | Pose Encoder | Fuser | 3D Decoder |
|---|---|---|---|
| conv2d(3,3,64,2) | fc(64) | fc(2048) | deconv3d(4,4,4,128,2) |
| conv2d(3,3,128,2) | fc(128) | fc(4096) | deconv3d(3,3,3,128,1) |
| conv2d(3,3,128,1) | - | - | deconv3d(4,4,4,64,2) |
| conv2d(3,3,256,2) | - | - | deconv3d(3,3,3,64,1) |
| conv2d(3,3,256,1) | - | - | deconv3d(4,4,4,32,2) |
| conv2d(3,3,256,2) | - | - | deconv3d(3,3,3,32,1) |
| conv2d(4,4,512,2) | - | - | deconv3d(4,4,4,32,2) |
| - | - | - | deconv3d(1,1,1,1,1) |

Table 2: Architecture of encoder, decoder fuser used in LSTM baseline.

**Depthnet for depth and mask estimation** The architecture of 2D U-net is composed of a encoder and decoder with skip connections. Skip connections join layer i(except for the last one) in encoder and layer n-i(except for the last one) in decoder where n is the total number of layers. The architectures of the encoder and decoder are shown in Table 3. We train 2D U-net using a L2 regression loss on depth and binary cross-entropy loss on mask. We regress on inverse depth rather than directly predicting the depth.

| Image Encoder(2D U-net) | Decoder(2D U-net) | Encoder(3D U-net) | Decoder(3D U-net) |
|---|---|---|---|
| conv2d(4,4,64,2) | deconv2d(4,4,512,2) | conv3d(4,4,4,16,2) | deconv3d(4,4,4,128,2) |
| conv2d(4,4,128,2) | deconv2d(4,4,512,2) | conv3d(4,4,4,32,2) | deconv3d(4,4,4,64,2) |
| conv2d(4,4,256,2) | deconv2d(4,4,512,2) | conv3d(4,4,4,64,2) | deconv3d(4,4,4,32,2) |
| conv2d(4,4,512,2) | deconv2d(4,4,256,2) | conv3d(4,4,4,128,2) | deconv3d(4,4,4,16,2) |
| conv2d(4,4,512,2) | deconv2d(4,4,128,2) | conv3d(4,4,4,256,2) | deconv3d(4,4,4,1,2) |
| conv2d(4,4,512,2) | deconv2d(4,4,64,2) | - | - |
| conv2d(4,4,512,2) | deconv2d(4,4,2,2) | - | - |

Table 3: Architecture of encoder and decoder in 2D U-net and 3D U-net.

Figure 2: **Improvement in 3D reconstruction IoU across views using different view selection policies.** *Top row:* Multi-object scenes. *Bottom row:* Single object scenes. The proposed active policy outperforms alternatives by choosing trajectories that "undo" occlusions.

Figure 3: Visualization of some camera trajectories selected by using active policy.

**3D reconstructor network** The architecture of 3D U-nets is similar to 2D U-net except that all the 2D conv and deconv layers are changed to 3D conv and deconv layers to match the 3D input and output. The architecture of encoder and decoder are shown in Table 3. We use sigmoid cross-entropy loss for training 3D U-net.

## 4 Visualizations

**Comparison between active policy and some baseline policies** As shown in Figure 2, percent increase in IoU between predicted and groundtruth voxel occupanices over single-view predictions are plot.

**Active trajectories** We show trajectories from both the *oneway* policy and the *active* policy in Figure 3.

**Multi-view reconstruction and 3D segmentation** Here we add more visualization results of reconstruction and 3D segmentation in Figure 4 and Figure 5. In Figure 6, we also included some results of 3D reconstruction on scenes sampled from SUNCG dataset. For data preparation on SUNCG dataset, We first split the house models provided in SUNCG into train and validation set.

Figure 4: Visualization results of 3D reconstruction on single and multiple objects using different methods.

Figure 5: Visualization results of 3D segmentation on multiple objects scene.

Then we sample scenes and voxel occupancy from those house models to create data for training and testing.

Figure 6: Visualization results of 3D reconstruction on sub-scenes sampled from SUNCG dataset.