[Reviews · NeurIPS 2018]

Reviewer 1



The paper proposes a method for reconstructing, segmenting and recognizing object instances in cluttered scenes in an active vision context, i.e. in situations where the camera can be moved actively by an agent, for instance a robot. Active vision is investigated actively in the recent past, and there seems to be a convergence between different communities, which start to develop similar methodologies for different objectives: (i) research in agent control, in particular with POMDP/Deep-RL, tackles realistic settings requiring visual recognition and includes geometry in its research, and (ii) research in vision/object recognition discovered Deep-RL to tackle active vision and creates neural/differentiable formulations of geometry based methods. The proposed method belongs to the second category and solves the problem with a 3D tensor representation, to which different subsequent views are unprojected and integrated. Depth and FG masks are estimated directly from RGB input using U-nets, and the different 3D representations are aligned with the first viewpoint in the sequence using ground truth odometry (using groundtruth ego-motion is standard in the community when working with simulated environments). From the 3D representation, which is updated using a recurrent layer, different output layers estimate different desired quantities: reconstruction map (voxel occupancy), segmentation map, etc. A policy output estimates the next view from a discrete action space. The method is very interesting and addresses many open problems, in particular how to perform these estimations from cluttered scenes on multiple objects jointly. Methodology-wise, it builds on existing work adding several key-components, in particular the 3D voxel-like representation which is updated with recurrent networks, as well as instance-wise output through clustering, which is responsible for the multiple-object capabilities. Here, we can remark that the number of objects in the scene needs to be known beforehand, since clustering is performed using k-means instead of a more sophisticated method capable of estimating the amount of instances. This is ok for the performed experiments on the ShapeNet dataset but seems to be an important restriction, unless some of the discovered instances can be estimated as no-objects using an additional confidence output (like in the 2D object detection literature like YOLO, SSD etc.). The paper is easy to read and to understand, and in large parts very easily understandable. However, that said, I have a couple of issues with the presentation and the lack of description of key details, which unfortunately concern the key differences to existing work. The paper is very dense and it is clearly difficult to present all the details, but the choices made here are not optimal. The first (and minor) part concerns the recurrent memory update, which is intuitively clear, but which lacks equations which related all key quantities to each other. The updates are described of being “recurrent”, but here is necessary to provide clearer indications on what is input, what is output, is there a dedicated hidden recurrent layer or does the 3D tensor play the role of the recurrent representation etc. It is, of course, not necessary to provide all the detailed equations of the different gates of a complex GRU cell; however, an RNN-like description is compact and can provide all relevant details, while it can simply be mentioned that GRU gates are part of the implementation. The second part concerns the way how the output is dealt with to estimate object instances. Firstly, an embedding tensor is mentioned but never introduced. It kind of becomes clearer in the following description, but this description is too short and not motivated. We don’t know the goal of this part of the paper, and we discover the objectives while reading it. The goal is to estimate outputs for each instance in the scene, which requires identifying the object instances first, done by clustering. This should be better motivated and related to the embedding tensor. Metric learning is thrown into the text in a quick way, referencing DRLIM (reference 11). In order to make the paper self-contained, this should be properly developed. Finally, all the different losses should be given correctly with equations. In particular, how does the method deal with the fact that multiple outputs are created for the scene (the different discovered object instances) and need to be matched to the multiple ground truth objects? In the 2D object detection literature, this is tackled in different ways, either using bi-partite graph matching / the Hungarian algorithm (Multi-box) or through a grid-based heuristic (YOLO, SSD). The experiments are convincing and show good performance against strong baselines. On a minor note, “random” and “oneway” are only shown in figure 3 and not figure 2. === After rebuttal === After rebuttal I keep my positive rating, since the rebuttal addressed some of my concerns. However, the following 4 references are very close to the paper and should be cited and properly referenced: - MapNet (CVPR 18): http://www.robots.ox.ac.uk/~joao/mapnet/ - IQA (CVPR 18): https://pjreddie.com/media/files/papers/IQA.pdf - Im2Pano3D (CVPR 18): https://arxiv.org/pdf/1712.04569.pdf - Neural Map (ICLR 2017): https://arxiv.org/abs/1702.08360

Reviewer 2



The author proposed semantic 3D reconstruction by learning a multi-view stereo machine and a next best view predictor simultaneously. As viewpoint suggestion is not differentiable, the authors used REINFORCE for training. It has been a fun read. I think this paper is well-written. Qualitatively the results look nice. The reference is thorough. The system is similar to LSM which was presented in last year’s NIPS, but the authors also takes next best view prediction into account. I’m on the border for this paper. My main concern is the lack of evaluation. While the system focuses on using neural nets to predict the next best view, its evaluation on viewpoint selection does not include other published methods. For example, how would the selected views compare with those selected by 3D ShapeNet (CVPR 15), as well as its follow-ups? Such a comparison is important to validate the main claim of the paper. Currently, all results are obtained on synthetic data. It’d be great if the authors can include some results on real data. Usually for real images camera poses might not be fully available. This will pose challenges to the proposed approach. Also, next best view selection is an important problem in robotics, where experiments on real data is crucial and heavily weighted. Minor: training with REINFORCE is often unstable. How often does active vision actually provides an advantage over random/one-way? It’d be good to include success rates and error bars. ==post rebuttal== The rebuttal partially addressed my concern. The comparison with one-step greedy view selection is good, but the results are not very impressive. The results on real datasets and a quantitative evaluation of REINFORCE are promised, but not included yet. I'm fine with accepting the paper, if what is promised can be included in the final version, but cannot champion it.

Reviewer 3



SUMMARY This paper addressed the problem of aggregating information from multiple camera views into a 3d tensor feature that keeps geometric consistency. The method uses the relative camera poses to align the unprojected feature tensors and combines them with GRU-based recurrent networks. This 3d tensor feature is used for instance segmentation, object classification, and voxel occupancy prediction. To select the most informative camera views, the authors used reinforcement learning (REINFORCE) to train a policy that selects the next adjacent view based on current states. TECHNICAL QUALITY I enjoyed reading this paper overall. Both the technical approach and the experimental results are clearly presented. The authors leveraged the structure of the active perception problem to bake in the 3D geometric consistency into the learned feature representations. This approach has outperformed previous approaches that do not take advantage of such geometric structure. Furthermore, it adopted a similar approach as in previous work, e.g., [14], to learn a policy to select best camera views. However, this approach presents several limitations as follow: 1. The approach works only in low resolution, which might constrain its scalability in more complex scenes. In the experiments, the 3D scenes are voxelized into a size of 64x64x64. Accordingly, the unprojected feature tensor has a size of 64x64x64x7, which contains more than 1 million elements. Increasing the resolution would lead to a larger model size and GPU memory consumption, and a longer computation time. Looking at Fig. 4, even the ground-truth voxels roughly approximate the 3d meshes of two objects. Such low-resolution approximation is likely to lead to notable issues in cluttered scenes with a larger number of objects. 2. This approach adopted a segmentation embedding method to obtain instance segmentation masks. It used k-means algorithm on these embeddings to group the voxels into k-sets. This seems to suggest that the model knows the number of objects in the scene a priori? How would this model work when the number of objects is large and unknown? Question about the technical details: When training the geometry-aware visual memory, what are the action sequences being used for selecting the camera views? Did you train the visual memory, and train the view selection policy, and then alternate? Or just using a random policy when training the visual memory? If so, would it perform better if the visual memory and the view selection policy are trained iteratively? EXPERIMENTAL VALIDATION The quantitative and qualitative results have shown that the model’s performance has improved as information from additional views is aggregated into the visual memory. Qualitatively, the new views selected by the RL policy look sensible for gathering informative geometric information of the scene. I am particularly impressed by the chair example in Fig. 2 where the model learned to reconstruct the thin arms of the chair after seeing two to three views. Questions about the experiments: 1. What is the intuition why the 3d reconstruction performance (Fig. 3) slightly dropped in the car category when the fourth view is added? 2. Can you also include curves of oracle view selection policy performance in Fig. 3, where the policy always selects the next camera view that improves the 3D reconstruction IoU the most (assuming known ground-truth)? This offers an insight into the upper bound performance of the visual memory independent from the view selection policy. OVERALL While some limitations of the proposed approach exist (as the authors also discussed in the last section of the paper), this work has offered a good attempt to learn 3D scene features with geometric consistency and has shown effective in simple domains for joint segmentation, classification, and reconstruction. My initial impression of this paper is positive.